# Deep Brain Stimulation and Hypoxemic Perinatal Encephalopathy: State of Art and Perspectives

**DOI:** 10.3390/life11060481

**Published:** 2021-05-25

**Authors:** Gaëtan Poulen, Emilie Chan-Seng, Emily Sanrey, Philippe Coubes

**Affiliations:** 1Unité “Pathologies Cérébrales Résistantes”, Department of Neurosurgery, Montpellier University Hospital, 34295 Montpellier, France; e-chan_seng@chu-montpellier.fr (E.C.-S.); e-sanrey@chu-montpellier.fr (E.S.); p-coubes@chu-montpellier.fr (P.C.); 2Unité de Recherche sur les Comportements et Mouvements Anormaux, Department of Neurosurgery, Montpellier University Hospital, 34295 Montpellier, France; 3Institut de Génomique Fonctionnelle, 34070 Montpellier, France; 4CNRS UMR5203, 34070 Montpellier, France; 5INSERM U661, 34070 Montpellier, France

**Keywords:** deep brain stimulation, internal globus pallidus, hypoxemic, anoxic, encephalopathy

## Abstract

Cerebral palsy (CP) is a heterogeneous group of non-progressive syndromes with lots of clinical variations due to the extent of brain damages and etiologies. CP is majorly defined by dystonia and spasticity. The treatment of acquired dystonia in CP is very difficult. Many pharmacological treatments have been tried and surgical treatment consists of deep brain stimulation (continuous electrical neuromodulation) of internal globus pallidus (GPi). A peculiar cause of CP is neonatal encephalopathy due to an anoxic event in the perinatal period. Many studies showed an improvement of dystonia in CP patients with bilateral GPi DBS. However, it remains a variability in the range of 1% to 50%. Published case-series concerned mainly small population with a majority of adult patients. Selection of patients according to the clinical pattern, to the brain lesions observed on classical imaging and to DTI is the key of a high success rate of DBS in children with perinatal hypoxemic encephalopathy. Only a large retrospective study with a high number of patients in a homogeneous pediatric population with a long-term follow-up or a prospective multicenter trial investigation could answer with a high degree of certitude of the real interest of this therapeutic in children with hypoxemic perinatal encephalopathy.

## 1. Introduction

Cerebral palsy (CP) is a heterogeneous group of non-progressive syndromes with lots of clinical variations due to the extent of brain damages and etiologies. CP is majorly defined by dystonia and spasticity. The prevalence of CP is estimated with 2–3 births per 1000 [1]. The Taskforce of Childhood Movement Disorders defines dystonia as a disorder in which involuntary sustained or intermittent muscle contractions cause twisting and repetitive movements, abnormal postures or both [2]. Disruption of motor control is always reported in CP. Cerebral palsy remains imprecise and Rosenbaum P et al. concluded in 2007, that, despite 150 years of debate, there was not a universally accepted definition of CP [3]. Movement disorders in this pediatric population have a severe impact on the growing of the musculoskeletal system leading to irreversible skeletal deformations [4].

The treatment of acquired dystonia in CP is very difficult. Many pharmacological treatments have been tried such as antidopaminergic, anticholinergic agents, benzodiazepines, baclofen (oral or intrathecal). Surgical treatment consisting of deep brain stimulation (continuous electrical neuromodulation) of internal globus pallidus (GPi) have shown divergent results. New strategies should be developed.

A peculiar cause of CP is neonatal encephalopathy due to an anoxic event in the perinatal period. It results in cerebral damages leading to a motor disorganization. Damages observed on MRI are variable. Global cortical atrophy, central atrophy, cerebellar atrophy, putaminal and GPi or thalamus atrophy can be observed. Many lesions can be described in the central grey nuclei and in the thalamus, highlighted by heterogeneous hypersignal on T2 MRI. Until now, no correlation has been demonstrated between the degree of lesions on MRI, the severity of the motor symptoms and the success rate of DBS.

GPi DBS is nowadays widely used in movement disorders patients. Since 1996, we demonstrated efficacy and safety of this surgical treatment, initially for patients with “primary” dystono-dyskinetic syndromes in children and in adults [5]. Consecutively, surgical indications extended to the “secondary” DDS. Although, the efficacy of GPi DBS has been demonstrated in selected cases of hypoxic neonatal encephalopathy, the precise mechanisms of action remain unclear.

## 2. State of Art

Elia AE et al. [6] published a review of literature assessing the efficacy of DBS for dystonia due to CP. They included many types of CP and not only patients with hypoxic perinatal encephalopathy. Despite these inclusion criteria, only 124 patients divided into 12 studies were included between 2009 and 2017 demonstrating the lack of data concerning this population of patients. However, beyond the medical aspect, socio-economic impact of children with multiple disabilities is major. There is currently a real need for effective treatment for these patients. The major difficulty remains the perfect selection of patient accessible to DBS. Nevertheless, Elia AE et al. reported a global reduction in motor symptoms of dystonia in all studies with bilateral GPi DBS. Firstly, Vidailhet et al. reported in a multicenter prospective pilot study that bilateral GPi DBS in 13 adults suffering from dystonia-choreoathetosis CP permitted an improvement of motor score (BFMDS) at 1 year and an improvement of pain, disability, and quality of life [7]. Their population included patients with CP secondary to neonatal hypoxic. However, the small size of the population did not permit to highlight the potential clinical factors that could be predictive of a better motor outcome after GPi-DBS. Furthermore, they implanted only adults. No study with a high number of children early implanted with bilateral GPi DBS has been reported in CP secondary to neonatal hypoxia.

Koy et al. performed a meta-analysis of the effects of DBS in dyskinetic CP. They reported a global improvement of BFMDRS at 1 year [8]. However, their data were heterogeneous, coming from 20 articles including 68 patients. In another study, Koy A. et al confirmed the effects of bilateral GPi DBS in dyskinetic CP on a population of eight adult patients [9]. Six of them suffered from a peripartal asphyxia. Despite a good improvement of dyskinetic symptoms, no objective improvement of dystonia, gait, speech, and swallowing were demonstrated.

In 2018, Fehlings D et al. reported on 13 DBS studies a reduction in dystonia in half of them. Four of 12 failed to demonstrate an improvement of dystonia [10]. Limitations of these studies are the lack of homogeneity of clinical subtypes of patients with CP in whom surgery is anticipated (constantly mixing together choreoathetosis, dyskinetic, extrapyramidal form) and, also, the lack of precise description of anatomical sequellae based on MRI. Heterogeneity of studied population and the limited number of patients remain the main limitations.

Keen JR et al. reported a retrospective study including 5 children with dystonic CP treated by bilateral GPi DBS [11]. Among these 5 children, only 2 suffered from a hypoxic encephalopathy. They highlighted a modest result concerning BADS and BFMDRS. Olaya JE et al. reported on a retrospective series of 9 patients implanted for secondary dystonia, a discrepancy between the absence of significant difference using usual motor scales (BFMDRS and BADS) and the subjective benefit observed by patients, families, or caregivers [12]. The actual system of assessment of this complex and mixed clinical syndrome is constantly described as being unsuitable.

Kim JP et al. compared on 10 adult patients with CP, the efficacy of bilateral GPi DBS versus Gpi DBS plus Ventralis Oralis thalamotomy [13]. They included only patients with fixed deformities. They demonstrated no clear improvements concerning the overall BFMDRS between the 2 groups. Romito LM et al. reported on 15 patients with GPi DBS for dystonia due to CP with an improvement of motor symptoms starting from 2 years after implant [14]. In addition, they found no difference between patients with or without brain lesions and between patients with or without spasticity concerning the motor outcome. This will be discussed.

Lin S. et al [15] reported a case of a 21-year-old women with severe fixed dystonia associated with spasticity with a story of birth hypoxia. First, she benefited from a surgical treatment with GPi DBS. Results of GPI DBS was insufficient. Then, she benefited from a high frequency DBS of superior cerebellar peduncles (SCP). At 6-months follow-up, they described an improvement of 36.4% of the BFMDRS and a significant improvement in her quality of life evaluated by the SF-36. They observed also a better modified Ashworth scale after surgery. Data are summarized in Table 1. Rose Davis et al. [16] and Galanda et al. [17,18,19] have also reported good results on spasticity in CP with stimulation of cerebellar anterior lobe. Although, international experience with GPi DBS in DDS demonstrated a very significant efficacy on the hyperkinetic component of movement disorder, it has been shown that a prominent hypertonic dystonia is not corrected by the neuromodulation of GPi [20]. This condition is quite constantly associated to spasticity explaining the limited impact of pre-pyramidal neuromodulation. In this case, cerebellum and, in particular, dentate nucleus could be a target. Indeed the desynchronization coming from denta-rubro-thalamic tract dysfunction and the potential direct action on the dentato-spinal tract might be adjusted. Dysfunctional cerebellar outputs can lead to movement disorders.

In addition to the motor improvement by DBS in CP, Perides S et al. [21] demonstrated that DBS decrease significantly dystonic pain in children with CP. Given that pain plays an important role in quality of life of these children, it can be an important goal for DBS surgery in CP, and a main criterion for selection.

Another important and underestimated factor, susceptible to explain the heterogeneity of results between series is the surgical technique: many surgical procedures are used for the targeting: anatomical methods using atlas, MRI-aligned frame-based, frameless targeting, or mixed anatomical and functional methods with a combination of MRI-aligned targeting and microelectrode recording. In our team, we use a MR imaging-based surgery under general anesthesia, without microelectrode recording, with a high degree of accuracy and reproducibility and a low rate of complications [22,23,24]. This is particularly adapted to the pediatric population.

## 3. Limitations

Even if many studies showed an improvement of dystonia in CP patients with bilateral GPi DBS, it remains a variability in the range of 1% to 50% [6]. An appropriate selection of patients should permit a better final DBS efficacy. The therapeutic success of bilateral GPi DBS is better in patients with prominent dystonic symptoms. Published case-series concerned, mainly, a small population with a majority of adult patients. Data on DBS in a pediatric population are rare. In addition, there are no data concerning the long term assessment of the implanted patients.

Prognostic factors of a better improvement by DBS are not well known yet. Some factors might have an impact on the clinical outcome as the younger age of implantation, the absence of fixed skeletal deformities, the absence of brain lesions on the MRI, the degree of spasticity and the clinical form of dystonia. The lack of knowledge about the pathophysiology of acquired dystonia can explain heterogeneity of clinical results of DBS. Target choices are usually guided by the pathophysiology. In addition, variations of stimulation settings can affect the efficacy of DBS.

In literature, the BFMDRS score is commonly used to assess outcomes after DBS. However, this score has been developed and validated for adult patients with primary dystonia [25]. The evaluation of children with hypoxemic perinatal encephalopathy is most complex, and BFMDRS and BADS are not enough to assess accurately the clinical outcome of these patients. Indeed, these children presented often a complex motor disorder mixing dystonia, choreoathetosis, or spasticity. Furthermore, the assessment of subjective quality of life is difficult in this population. A precise assessment of motor function in daily life activities might be important but difficult to quantify. All data coming from these different studies are preliminary evidence of the potential interest of GPi DBS in CP children. However, only a large retrospective study with a high number of patients in a homogeneous pediatric population with a long-term follow-up or a prospective multicenter trial investigation could answer with a high degree of certitude of the real interest of this therapeutic in children with hypoxemic perinatal encephalopathy. The major concern remains the pre-operative selection of patients. Moreover, the effects of continuous neuromodulation on the maturating brain remains unknown yet.

## 4. Perspectives

Until now, DBS was proposed only for children with severe forms of movement disorders resulting from a hypoxemic perinatal encephalopathy with no other possible therapy because of the uncertain results, the risk of infection, and the complexity of the procedure. However, with the low rate of complications of this procedure, the question of whether to implant children with a moderate form needs to be asked. Even if the treatment is invasive, it has been demonstrated that DBS is a safe and often used technique, and it remains reversible. Indeed, in the case of a lack of therapeutic effects, the stimulation equipment can be removed.

Functional imaging and diffusion tensor imaging (DTI) may be able to help to highlight abnormal patterns of neural activity in these children, and may help to select patients who would be good responders to the continuous neuromodulation.

Selection of patients according to the clinical pattern, to the brain lesions observed on classical imaging and to DTI is the key of a high success rate of DBS in children with perinatal hypoxemic encephalopathy. Further studies concerning functional connectivity and quantitative neurophysiological measurements could highlight phenotypic differences within this population and provide prognostic factors [26].

## 5. Conclusions

The major difference between patients with primary movement disorders and anoxic ones is the existence of anatomical lesions.

The dyskinetic component of the disorder is frequently associated to lesions in the grey nuclei but sparing cortical and sub-cortical connections. When severe, anoxia impaired the cortex and especially the central area. As a result, any attempt to improve pre-cortical movement regulation will fail because the pyramidal tract itself is not able to achieve the movement. It is the reason why the analysis on MRI of the different patterns and subtypes of lesions due to perinatal anoxia is crucial. Extended anatomical lesions on MRI are possibly a limiting factor to DBS.

Clinically, the existence of a motor deficit consecutive to pyramidal cortex and tract impairment is potentially a limiting factor to DBS efficacy. Another limiting factor should be the presence of a high degree of spasticity.

The indication of a treatment by DBS should be individualized based on motor symptoms, dystonia, the severity of spasticity, and the anatomical lesions on MRI.

To conclude, recent advances in the management of patients with movement disorders due to a perinatal hypoxia should not be grouped in the category of CP but identified by the etiology of the disease and considered separately. 

## Figures and Tables

**Table 1 life-11-00481-t001:** Studies assessing bilateral GPi (associated or not with other targets) DBS for patients with dystono-dyskinetic syndromes due to a cerebral palsy.

Study	Design	Number of Patients	Target	Mean Age at Surgery (Years)	Follow-up (Months)	Main Results
Vidailhet et al. 2009 [7]	Prospective study	13	GPi	32.6 (range 20–44)	12	BFMDRS-M: −24.3% BFMDRS-D: −13.3% SF-36:15% SCL-90: −34.9%
Kim JP et al. 2012 [13]	Case series	10	GPi versus GPi plus Ventralis Oralis thalamotomy	26.8 (range 18–37)	31.8 (range 12–86)	GPi: BFMDRS-M: −32% BFMDRS-D: −14.3% GPi + ventralis oralis thalamotomy: BFMDRS-M: −31.5% BFMDRS-D: −0.18%
Koy et al. 2013 [8]	Meta-analysis	68 (20 studies)	GPi	-	-	BFMDRS-M: −23.6% BFMDRS-D: −9.2%
Olaya JE et al. 2013 [12]	Case series	9	Gpi (8) GPi and STN (1 patient)	16.3 (range 6–20)	3.8 (range 0.5–9)	BFMDRS: −10.5% BADS: −7.9%
Koy et al. 2014 [9]	Case series	8	GPi	26.1 (range 16.1–33.8)	44.5 (range 8–83)	BFMDRS-M: −1.2%
Keen JR et al. 2014 [11]	Case series	5 (2 with hypoxic encephalopathy)	GPi	11.4 (range 8–17)	26.6 (range 2–42)	BFMDRS-M: −27.9% BADS: −16.3%
Romito LM et al. 2014 [14]	Prospective	15	GPi	29.8 (range 15–47)	52.8 (range 24–84)	BFMDRS: −49.2% BFMDRS-D: −30.4% SF-36: 44.8%
Elia AE et al. 2017 [6]	Review (12 studies)	124	GPi	-	2–132	BFMDRS-M: −23.6% BFMDRS-D: −9.2%
Fehlings et al. 2018 [10]	Review	(13 studies)	-	-	-	Reduction of dystonia in six studies
Lin S et al. 2020 [15]	Case report	1	GPi and both superior cerebellar peduncles and dentate nuclei	21	6	BFMDRS-M: −36.4% BFMDRS-D:−33.3%

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
