# Peer review of "Deep Brain Stimulation and Hypoxemic Perinatal Encephalopathy: State of Art and Perspectives"

_life, 2021, doi:10.3390/life11060481_

Round 1
Reviewer 1 Report
Excellent review of the literature. important topic. agree with the authors overall conclusion that the results of GPi DBS are modest especially when the is marked abnormality on MRI however disagree with the conclusion that a trial of DBS is warranted because there are no other treatments and DBS is reversible and can be removed. Perhaps a conclusion that cases should be individualized based on motor symptoms/ degree of spasticity vs dystonia, extent of MRI changes in different grey/white matter would add more depth to this submission.
Reviewer 2 Report
This review paper by Poulen et al. summarizes the state-of-the-art knowledge about the outcomes of deep brain stimulation for cerebral palsy due to perinatal hypoxia. The authors have selected a very interesting, evolving topic that is of interest to a broad audience of neurosurgeons and neurologists. While the manuscript covers the published cases exhaustively and with an appropriate amount of detail on clinical outcome, a summary table showing the reported cases is sorely missing. It makes is much harder for the reader to grasp and retain the message. After all the earnest effort the authors put into this report, the manuscript should not get published without a summary table. Also, English language proofing is needed; several sentences are confusing (e.g., “The final functional result is directly reliable to an appropriate selection of patients”).
